# Is Full-Time Equivalent an Appropriate Measure to Assess L1 and L2 Perception of L2 Speakers with Limited L2 Experience?

Celia Gorba

Facultat de Filosofia i Lletres, Universitat Autònoma de Barcelona, 08193 Bellaterra, Spain; celia.gorba@uab.cat

**Abstract:** The revised version of the Speech Learning Model (SLM-r) regards full-time equivalent (FTE), which involves the amount of L2 use during the length of residence (LOR) in an L2 setting, as the main factor in L2 speech acquisition. Previous studiesshowed that LOR has a significant effect on L2 and L1 production and perception but does not explain differences between populations (i.e., L1-Spanish L2-English vs. L1-English L2-Spanish). A reanalysis of the data has been conducted by calculating the FTE of the experienced participants. The aim was also to investigate whether the assumptions of the SLM-r are applicable to L1 and L2 perception. A series of correlation tests between FTE and category boundary—between voiced and voiceless stops—was conducted, yielding non-significant results. The relatively short LOR of participants, the quality of the input and differences in terms of L2 instruction between participants could explain the lack of a clear effect of FTE in this study. Therefore, FTE on its own may not be sufficient to account for L2 accuracy in perception, at least for L2 speakers with limited L2 input, and other factors should be considered.

**Keywords:** perception; L2 acquisition; L1; crosslinguistic influence; SLM; FTE

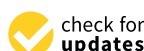



## 1. Introduction

The main factors that modulate crosslinguistic influence (CLI) have been widely investigated, including length of residence in an L2 setting (LOR), age of arrival, language dominance, L2 use, and L2 instruction, amongst others (e.g., Piske et al. 2001). The most widely cited L2 speech models, such as the Speech Learning Model (SLM, Flege 1995) or the Perceptual Assimilation Model (PAM, Best 1995; Best and Tyler 2007) traditionally regarded L2 experience, quantified mostly as LOR, as the most relevant factor. More recently, the revised version of the SLM (SLM-r, Flege and Bohn 2021) proposed Full-Time Equivalent (FTE), which comprises both LOR and L2 use, as an alternative to quantify L2 experience. The present study consists of a reanalysis of the perceptual data presented in Gorba (2019, 2020), which quantified L2 experience in terms of LOR, with FTE as the main factor. Its goal is to determine whether FTE values for L2 speakers with limited experience can account for differences in the perception of L2 and L1 stops in two parallel populations, namely English learners of Spanish and Spanish learners of English with varying experience with the L2. The main contribution of the present study lies in the fact that it tests a newly proposed measure, FTE, on the perception of not only the L2 but also the L1 in two different populations. Specifically, the perception of the voicing contrasts /p/-/b/ and /k/-/g/ along a voice-onset time (VOT)—i.e., the time between the release of the stop and the onset of voicing—continuum was tested. Coronal stops were not included because they are produced with a different place of articulation in the two languages investigated (i.e., they are alveolar in English and dental in Spanish).

Before focusing on FTE, it is pertinent to briefly go over the concept of language experience. L2 experience has been given a number of different definitions in the literature but has mainly been quantified as LOR in previous studies (e.g., Flege 1987, 1995; Flege et al. 1997; Gorba 2019; Gorba and Cebrian 2021; Jun and Cowie 1994; Levy and Law 2010). This factor has been found to modulate L2 production, as a longer LOR tends to

result in a more target-like performance in the L2 (e.g., Flege et al. 1997; Lev-Ari and Peperkamp 2013; Levy and Law 2010). As for the L1, having the experience of living in an L2 setting has been reported to result in less target-like L1 productions or perceptions influenced by the L2 (Cebrian 2006; Dmitrieva 2019; Flege 1987; Kartushina et al. 2016; De Leeuw et al. 2010, 2012; Major 2010). The widely cited study by Flege (1987) found a bidirectional influence on the production of /t/ on French learners of English living in the US. Note that, in initial position, English is a language that contrasts the presence of aspiration (long-lag VOT)—signaling voiceless stops—with its absence (short-lag VOT or, less frequently, voice-lead VOT)—signaling voiced stops. French, on the other hand, contrasts short-lag VOT in voiceless stops with voice-lead VOT—i.e., prevoicing—in voiced stops. Thus, VOT values in English are greater than in French. The French learners of English in Flege's study produced English /t/ with VOT values that were significantly shorter—more French-like—than those of English monolinguals, and French /t/ with significantly longer—more English-like—VOT values than French monolinguals. However, in some cases, no effect of the L2 on the L1 was observed in learners living in an L2 setting (e.g., Gorba 2016; Gorba and Cebrian 2023). It should be noted that these studies used LOR as the main factor, which was not found to be a successful predictor of L1 attrition in previous studies (De Leeuw et al. 2010). Moreover, previous research has not always found a linear relationship between LOR and L1 drift. Specifically, Chang (2012, 2013) found that L1-English beginner learners of Korean in an L2 setting presented more L2 influence on their L1 VOT than those who already had some knowledge of the L2, which was interpreted as a novelty effect that may decrease with L2 familiarity. As for the effect of experience of living in an L2 setting on L2 perception, similarly to production, some previous studies have found a beneficial effect on the L2, since a greater LOR tends to result in a more accurate perception (e.g., Bohn and Flege 1990; Flege et al. 1997). The effect of LOR on L1 perception has received less attention. Some previous studies found that LOR modulated the influence of the L2 on the L1—i.e., a greater LOR resulted in a greater influence of the L2 on the L1 (e.g., Cabrelli et al. 2019; Cebrian 2006; Major 2010)—whereas others did not (e.g., Gorba 2016; Gorba and Cebrian 2021), possibly due to a shorter LOR on the part of the subjects with experience in an L2 setting than in other studies (e.g., Cebrian 2006: 24 years vs. Gorba 2016: 1 year; Gorba and Cebrian 2021: 4 years).

To our knowledge, the only previous studies that investigated the effect of L2 experience —quantified as LOR—on the perception of L1 stops are previous studies by Gorba and colleagues (Gorba 2016, 2018, 2019, 2020; Gorba and Cebrian 2021). Gorba (2020) investigated the same participants as those tested in the current study (i.e., English learners of Spanish and Spanish learners of English differing in amount of L2 experience), as well as a group of L2 speakers with no experience in an L2 setting and a control group of monolinguals for each population (i.e., a total of eight groups). In this case, L2 speakers of each language were divided into three groups according to their amount of L2 experience. Note that, considering the VOT values of each language (see Section 1.1 for more details), English speakers present later category boundaries—i.e., the VOT value in which perceptual response switches from one category to another—than Spanish speakers, as the former use aspiration contrastively whereas the latter use prevoicing. In Gorba (2019, 2020) and Gorba and Cebrian (2021), different results were found for the English and the Spanish speakers. In the case of the L1-English learners of Spanish, all groups, independent of L2 experience, were found to perceive both L1 and L2 stops using English-like values. That is, there was an influence of the L1 on the L2 and no effect of the L2 was observed on the L1, as they perceived both languages using similar values which did not differ from one another. As for the Spanish speakers, there were differences between groups modulated by L2 experience. Only the least experienced learners were found to present significantly earlier category boundaries in their L2 than the English monolinguals, whereas the groups with experience living in an L2 setting did not. As for the L1, the most experienced learners had significantly later—i.e., more English-like—category boundaries than the Spanish monolinguals, suggesting an effect of the L2 on the L1. Regarding the differences between the L1

and the L2, in most cases, all Spanish groups presented similar—intermediate between the two languages—values in the L1 and in the L2, which did not differ from each other. The different outcomes between the two populations were explained by differences in terms of L2 use and amount of L2 instruction. Specifically, there were smaller differences between the English participants in the UK than in the case of the Spanish groups (a difference of only 4% for weekly use between the most and least experienced English participants and of 42% regarding the Spanish groups). Moreover, the Spanish participants had received a greater amount of L2 instruction than the English learners (13.4 vs. 2.4 years for the most experienced groups, respectively). This outcome suggests that not only length of residence, but also other factors are highly relevant to CLI in perception.

Some prior studies have also suggested that LOR on its own may not represent the quantity of the input received. Using an error detection task, Flege and MacKay (2004) examined the perception of English vowels by L1-Italian speakers living in Canada. Results showed that late learners who seldom used the L1—and thus used the L2 to a greater extent—and early learners who often used it performed similarly. In fact, increased L2 use—even in a non-L2 setting—can result in more target-like L2 productions and in L1 drift towards the L2, as revealed by Kartushina and Martin (2019) in a study in which English was used as a lingua franca for a relatively short period of time—i.e., a two-week study abroad program. This suggests that both length of residence and frequency of L2 use contribute to the quality of perception, such that frequent use can compensate for late exposure and vice versa. Thus, the revised version of the SLM (SLM-r, Flege and Bohn 2021) emphasizes the importance of individual variation and proposes a new variable as the most relevant one in L2 speech acquisition and CLI, namely FTE. It is a compound variable which is calculated by multiplying the proportion of L2 use by LOR. For example, Flege and Bohn (2021) calculated the FTE for the participants in Aoyama and Flege (2011), who were native Japanese learners of English living in the US. Most participants had lived in the L2 setting for less than 5 years, but L2 use varied across participants, as they reported using English from 0% to 97% of the time. Considering these two variables, FTE was calculated for each participant. Three groups with different amounts of FTE were obtained, namely a low-input group, with a mean of 0.09 years of FTE (range: 0–0.2), a mid-input group, with a mean of 1.01 FTE (range: 0.4–1.8), and a high-input group, with a mean of 5.46 FTE (range: 2.1–14.4). Participants were exposed to productions of English /ɹ/, /l/ and /w/ and of Japanese /R/ and /w/ and had to determine the degree of dissimilarity between the tokens and Japanese /R/ on a scale from 1 to 7. Results showed that FTE was correlated to the dissimilarity ratings for /ɹ/, but not for /l/, indicating that, as FTE increased, Japanese learners of English perceived /ɹ/, but not /l/, to be increasingly different from Japanese /R/. Moreover, Flege (2021) determined that at least 2 years of FTE were needed for the Japanese learners to perceive the differences between the L1 and L2 liquids, as only participants with higher input were able to do so. In sum, although LOR has been found to modulate CLI, other factors may play an important role. For this reason, the SLM-r proposed FTE, a more comprehensive variable that comprises not only LOR but also L2 use. Next, VOT, the feature analyzed in the current study, for English and Spanish plosives, will be reviewed.

### 1.1. Crosslinguistic Differences in VOT

The current study focuses on the perception of the voicing contrast of stops in English and in Spanish. Table 1 illustrates the mean VOT of stops in initial position for English and Spanish reported in previous studies. The voicing distinction in these two languages is cued mainly by VOT, but different values are used in each language. In English, voiceless stops are produced with long-lag VOT—which is perceived as aspiration—whereas voiced stops are normally produced with short-lag VOT, although voice-lead VOT may also be used in free variation (e.g., Lisker and Abramson 1964; see Table 1 for actual reported VOT values). According to Lisker and Abramson (1964), speakers of American English produced /p/ in stressed initial position with a VOT of 58 ms, /t/ with 70 ms and /k/ with 80 ms. Regarding

voiced stops, most participants used short-lag VOT, although some used voice-lead VOT (/b/: 1/−101 ms; /d/: 5/−101; /g/: 21/−88 ms). Similar results were reported by Docherty (1990), who examined British speakers. Voiceless stops were produced with long-lag values (/p/: 42 ms; /t/: 65 ms; /k/: 62 ms), and voiced stops were mostly produced with short-lag VOT (/b/: 15 ms; /d/: 21 ms; /g/: 27 ms). A few instances of prevoiced productions of voiced stops were found but discarded to calculate the mean.

Spanish presents a shorter VOT range, i.e., from voice-lead VOT for voiced stops to short-lag VOT for voiceless stops (see Table 1). Unlike in English, aspiration is not used in voiceless stops in Spanish, and prevoicing has a distinctive function. Lisker and Abramson (1964) reported that Puerto Rican speakers of Spanish produced /p/ in initial position with a mean VOT of 4 ms, /b/ with −138 ms, /t/ with 9 ms, /d/ with −110 ms, /k/ with 29 ms and /g/ with −108 ms. In a similar fashion, speakers of Castilian Spanish produced /p/ with a mean VOT of 6.5 ms, /b/ with −69.8 ms, /t/ with 10 ms, /d/ with −78 ms, /k/ with 25.7 ms and /g/ with −58 ms (Castañeda 1986).

**Table 1.** Mean VOT values reported for English and Spanish stops by previous studies (SD = standard deviation; r = range).

| Language/Study | /p/ | /b/ | /t/ | /d/ | /k/ | /g/ |
|---|---|---|---|---|---|---|
| **English** | | | | | | |
| -British English (Docherty 1990) | 42 (SD: 10) | 15 (SD: 8) | 65 (SD: 13) | 21 (SD: 6) | 62 (SD:14) | 27 (SD: 11) |
| -American English (Lisker and Abramson 1964) | 58 (r: 20–120) | 1/−101 (r: 0–5/ −130– −20) | 70 (r: 30– -105) | 5/−102 (r: 0–25/ −155– −40) | 80 (r: 50– −135) | 21/−88 (r: 0–35/ −150– −60) |
| **Spanish** | | | | | | |
| -Puerto Rican Spanish (Lisker and Abramson 1964) | 4 (r: 0–15) | −138 (r: −235– −60) | 9 (r: 0–15) | −110 (r: −170– −75) | 29 (r: 15–55) | −108 (r: −165– −45) |
| -Castilian Spanish (Castañeda 1986) | 7 (SD: 6) | −70 (SD: 25) | 10 (SD:5) | −78 (SD: 26) | 26 (SD: 11) | −58 (SD: 26) |

## 2. Materials and Methods

### 2.1. Subjects

A total of 37 participants completed this study, namely 19 English learners of Spanish and 18 Spanish learners of English. All participants were young adults aged between 18 and 32. The English learners of Spanish had lived in an L2 setting between 0.5 and 6 years, that is, an average of 2.1 years (SD: 2.1), used Spanish for an average of 16% (SD: 8%) of the time on a weekly basis and had learnt Spanish for an average of 5 years (SD: 4.9). Ten of them were living in Barcelona (Spain) at the time of testing and were English teachers, and none of them reported knowledge or use of Catalan. They were tested in two different institutions in the same area, namely Universitat Autònoma de Barcelona and Universitat de Barcelona. The remaining participants were university students who were majoring in Spanish studies—and thus were receiving formal instruction in Spanish—and had lived in a Spanish-speaking country (for at least six months) during the academic year prior to the experiment but were back in England at the time of testing. They were tested at the phonetics laboratory at Queen Mary University of London (QMUL). Although, particularly in the case of the English learners of Spanish living in Barcelona, there were some dialectal differences between participants, these were not found to have an effect on their perception (or production, as revealed by Gorba and Cebrian 2021) of L2 and L1 stops. As reported in Gorba and Cebrian (2021), the subset of participants majoring in Spanish had lived in an L2 setting for an average of 9 (SD: 4) months and had received 7.5 (SD: 5) years of formal instruction, whereas the English teachers in Spain had lived in an L2 setting for a longer period of time, about 4 years (51 months, SD: 27), but had received less formal instruction—i.e., 2.4 years (SD: 2). The average of L2 use, however, was similar for both the Spanish majors and the English teachers (15%, SD: 7; 16%, SD:11, respectively). Regarding the Spanish participants, they had lived in an English-speaking

country between 0.3 and 8 years, for an average of 2 years (SD: 2.2), used English an average of 39% of the time on a weekly basis (SD: 23%) and had studied English for an average of 13.7 years (SD: 2). Eight Spanish learners of English were professionals living in London at the time of the experiment, and they were also tested at QMUL. The remaining ten were students majoring in English Studies who had spent at least one term living in an English-speaking country during the academic year prior to the experiment; these participants were tested in the same Spanish institutions as the English teachers living in Spain. The amount of L2 instruction was very similar across the Spanish participants (an average of 13 years for both, SD: 2, for both), possibly because English is mandatory in the Spanish school curriculum. As for L2 use, the professionals in London used the L2 more than the English majors on a weekly basis (55%, SD: 25; 26%, SD: 10, respectively).

*2.2. Stimuli*

Two separate VOT continua, one for the bilabial contrast and one for the velar contrast, were created. Each continuum had 17 stimuli ranging from −30.4 ms to 57.9 ms and from −25 ms to 72 ms of VOT, respectively. The VOT range was decided considering the results obtained in a pilot study completed by 6 native speakers of each language—i.e., Spanish and English. Sufficient stimuli were provided to show clear crossover areas between the voiced and voiceless categories as well as unequivocal responses. Moreover, an equal number of stimuli were selected for each continuum, and the resulting tests were intentionally not made too long to avoid fatigue on the part of the participants. The stimuli presented stops followed by /i/, which is the most perceptually similar vowel across the two languages under study (Cebrian 2019). The continua were created manually by editing natural speech produced by a highly proficient L1-Spanish L2-English speaker who was a trained phonetician. The computer software Praat (Boersma and Weenink 2016) was used for the editing. Other voicing cues including burst intensity and duration and vowel formant transitions were controlled. Specifically, an ambiguous burst and an ambiguous vowel in terms of voicing were created by selecting the most appropriate production and modifying their duration and intensity (in the case of the burst) and the F1 and F0 contours of the vowel so as to obtain values that were intermediate between voiced and voiceless productions. The prevoiced stimuli were created by adding cycles of prevoicing extracted from the same production before the burst in steps of around 5 ms of VOT. Stimuli with positive VOT were created by adding chunks of aspiration in steps of about 5 ms between the burst and the vowel. The stimuli were piloted on 5 functional monolingual speakers of each language and deemed appropriate (see Gorba and Cebrian 2021 for all the values in the continua).

*2.3. Tasks*

Each continuum was presented separately in a forced-choice two-alternative identification task implemented in Praat in two different conditions, in a task presented in English and a task presented in Spanish. That is, the same stimuli were used to test the L1 and the L2 using different response alternatives according to the language that was being tested. For English, the response alternatives were "'b' as in beetle" and "'p as in peeler" for the bilabial continuum and "'g' as in geeky" and "'k' as in keeper" for the velar continuum. In Spanish, the response alternatives used for the /p/−/b/ and /k/−/g/ contrasts, respectively, were "'b' como en bicho" ("'b' as in bug") and "'p' como en pico" ("'p' as in peak") and "'g' como en guiso" ("'g' as in stew") and "'k' como en quito" ("'k' as in I remove"). The order of testing of language and place of articulation was counterbalanced. Participants also completed a reading task—the results of which are beyond the scope of the current paper—in each language prior to completing each perception test, which helped set the desired language mode. There was no break between the testing of the two languages, but participants watched a short TED Talk in the language that was going to be tested next to also help set the right language mode. Participants completed the tests in the phonetics laboratories located at the universities where they were recruited. In all testing locations, the same high-quality noise-cancelling headsets (Beyerdynamic DT 770 M) were used.

*2.4. Data Analysis*

Following the methodology used in previous studies (Gorba 2019; Gorba and Cebrian 2021), the measure that was used to quantify perception was category boundary, which was calculated for each participant in each language and contrast (/p/–/b/ and /k/–/g/). First, the total number of times a stimulus in the continuum was perceived as the voiceless category (/p/ or /b/) was calculated. Then, a logistic function was obtained for each participant using the computer software SPSS Statistics. The constant (b0) and slope (b1) of the resulting logistic function were then used in the formula -LN(b0)/LN(b1), resulting in a numerical value that represented the category boundary. The factor under study, FTE, was calculated following Flege and Bohn (2021). The number of years that each participant had lived in an L2 setting was multiplied by the proportion of L2 use on a weekly basis.

A series of one-tailed Pearson correlation tests (variables: FTE and category boundary) was conducted using SPSS. Specifically, a test was carried out for each population (L1-English and L1-Spanish), language (English and Spanish) and place of articulation (bilabial and velar), resulting in a total of 8 statistical tests. Given that four correlational analyses were carried out simultaneously for each population—one per language and place of articulation—$\alpha$ values were adjusted accordingly, following the Bonferroni method. Thus, only results with a *p* value lower than 0.0125 (i.e., the corrected *p* value) were considered to be significant. An individual analysis was also carried out. Moreover, the degree to which L2 use, LOR and L2 instruction predicted category boundary placement was also assessed. Specifically, a multiple regression analysis for each language, place of articulation and population was conducted. Initially, all variables (i.e., FTE, L2 use, LOR and L2 instruction) were considered for the analysis. However, multicollinearity, which has an effect on the *p*-value and *F* statistics (Freund et al. 2006; Sen and Srivastava 1990), was found to affect FTE. As expected, FTE, which is calculated by multiplying LOR by L2 use, was found to be highly correlated with the variables that make it up and, thus, its inclusion in the analysis would have been redundant. Multicollinearity was assessed by conducting the Durbin–Watson test for autocorrelation and a collinearity diagnosis (considering both tolerance and variance inflation factors). Thus, FTE was not included in the series of multiple regression analyses but was assessed by means of separate correlations as well as an individual exploration of the data. Before conducting the analyses (independent variables: LOR, L2 use and L2 instruction; dependent variable: category boundary placement for each place of articulation), the data were normalized by calculating the Z-scores for each independent variable. All mathematical assumptions were met, including the absence of multicollinearity.

The following sections present the results for each population. First, the results obtained by the L1-English learners of Spanish will be presented, then the results obtained by the L1-Spanish speakers.

## 3. English Learners of Spanish

*3.1. Results*

After following the procedure explained above, the resulting category boundaries and FTE for the English learners of Spanish were calculated (see Table 2 for all results). Overall, participants presented low to mid FTE values (range: 0–1.9), following Flege's (2021) classification, which indicates a limited amount of L2 experience. It should be noted that only five participants presented a value over 0.4—that is, an intermediate FTE value according to Flege's (2021) classification—only three were higher than 0.5, and only two of them had values close to 2, i.e., close to a high FTE. The scatterplots (Figures 1 and 2) also illustrated that most participants presented low FTE values, whereas only two of them presented values close to 2. The mean FTE was 0.4 (SD: 0.6) and the median was 0.2. As for the category boundaries, the calculated means were 16.1 for English /p/–/b/ (SD: 3.3; median: 17.4); 28.4 for English /k/–/g/ (SD: 4.2; median: 28.8); 15.7 for Spanish /p/–/b/ (SD: 4.1; median: 15.8) and 26.7 for Spanish /k/–/g/ (SD: 4.3; median: 27). The category boundaries calculated for each participant were correlated with their corresponding FTE.

This calculation was done for each language and place of articulation. None of the tests revealed a significant correlation (see Table 3 for the results).

**Table 2.** Percentage of L2 use, years spent in an L2 setting, FTE and the category boundaries for the bilabial and velar contrasts in English and Spanish by the English learners of Spanish.

| Participant | % Weekly L2 Use | Years in L2 Setting | FTE | English /p/–/b/ Boundary | English /k/–/g/ Boundary | Spanish /p/–/b/ Boundary | Spanish /k/–/g/ Boundary |
|---|---|---|---|---|---|---|---|
| En01 | 37% | 5.0 | 1.9 | 17.4 | 28.2 | 11.4 | 22.7 |
| En02 | 7% | 1.0 | 0.1 | 12.0 | 24.5 | 9.7 | 30.7 |
| En03 | 17% | 6.0 | 1.0 | 13.2 | 29.6 | 15.8 | 27.6 |
| En04 | 13% | 3.0 | 0.4 | 17.7 | 28.5 | 17.4 | 26.6 |
| En05 | 7% | 6.0 | 0.4 | 13.6 | 28.8 | 13.5 | 28.8 |
| En06 | 10% | 1.0 | 0.1 | 19.7 | 28.9 | 13.8 | 26.2 |
| En07 | 10% | 3.0 | 0.3 | 17.7 | 38.2 | 19.1 | 27.0 |
| En08 | 30% | 6.0 | 1.8 | 17.4 | 28.8 | 19.1 | 28.8 |
| En09 | 17% | 1.0 | 0.2 | 17.4 | 31.9 | 13.5 | 33.1 |
| En10 | 30% | 1.0 | 0.3 | 14.5 | 25.6 | 9.8 | 21.9 |
| En11 | 10% | 1.0 | 0.1 | 11.4 | 22.0 | 17.7 | 18.4 |
| En12 | 13% | 0.8 | 0.1 | 13.9 | 24.8 | 7.7 | 23.6 |
| En13 | 20% | 0.5 | 0.1 | 14.0 | 25.0 | 17.1 | 27.0 |
| En14 | 13% | 0.8 | 0.1 | 13.9 | 24.8 | 21.2 | 25.0 |
| En15 | 10% | 0.5 | 0.1 | 25.5 | 38.1 | 22.8 | 29.8 |
| En16 | 10% | 1.5 | 0.2 | 17.4 | 28.8 | 15.5 | 28.8 |
| En17 | 13% | 0.3 | 0.0 | 17.4 | 28.8 | 17.4 | 17.7 |
| En18 | 20% | 1.0 | 0.2 | 17.7 | 29.2 | 19.7 | 33.1 |
| En19 | 10% | 0.5 | 0.1 | 13.5 | 24.5 | 15.8 | 30.7 |

**Table 3.** Results for the correlation tests between FTE and category boundaries for the English learners of Spanish.

| | | English /p/–/b/ | English /k/–/g/ | Spanish /p/–/b/ | Spanish /k/–/g/ |
|---|---|---|---|---|---|
| FTE | Pearson Correlation | 0.051 | 0.078 | −0.057 | −0.038 |
| | Sig. (1-tailed) | $p = 0.418$ | $p = 0.375$ | $p = 0.408$ | $p = 0.438$ |
| | $N$ | 19 | 19 | 19 | 19 |

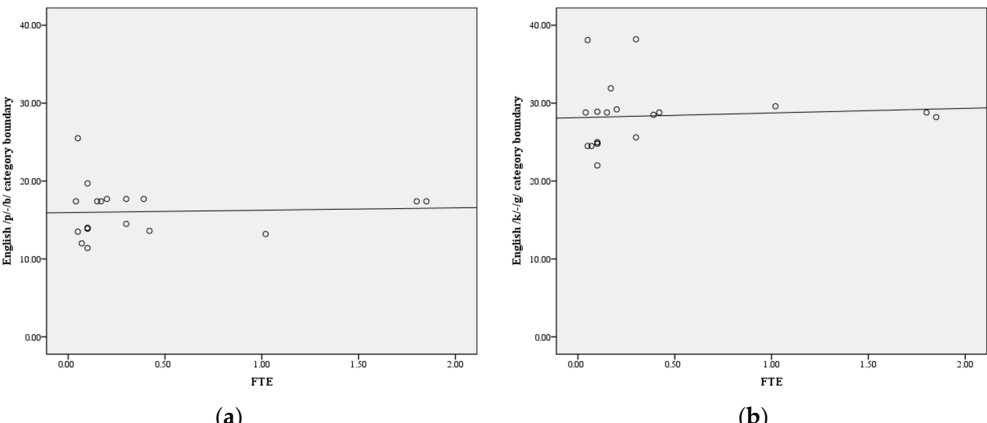

(a)　　　　　　　　　　　　　　　　　　(b)

**Figure 1.** Scatterplots with fit lines for FTE (*x* axis) and category boundary (*y* axis) in English by the English learners of Spanish. (**a**) Category boundary for the /p/–/b/ contrast in English. (**b**) Category boundary for the /k/–/g/ contrast in English.

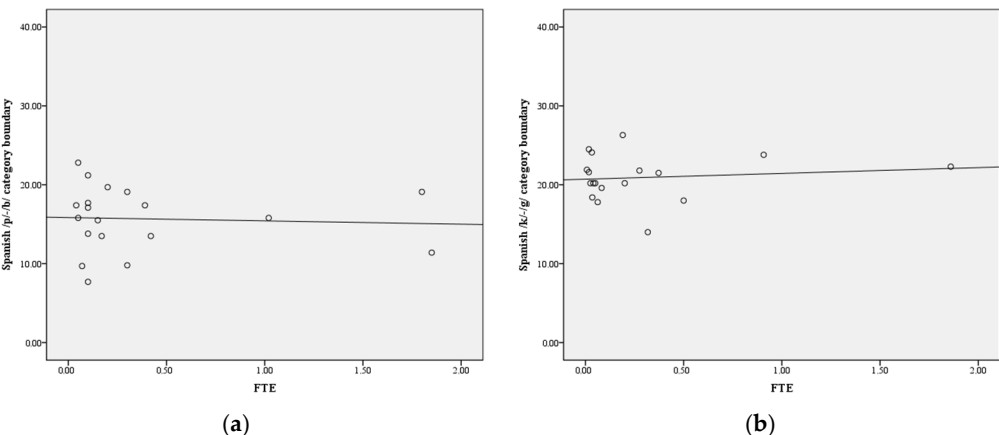

**Figure 2.** Scatterplots with fit lines for FTE (*x* axis) and category boundary (*y* axis) in Spanish by the English learners of Spanish. (**a**) Category boundary for the /p/–/b/ contrast in Spanish. (**b**) Category boundary for the /k/–/g/ contrast in Spanish.

Furthermore, the degree to which L2 use, LOR and L2 instruction (independent variables) predicted category boundary location (dependent variable) was assessed by means of a series of multiple regression analyses—i.e., one per contrast. The statistical analyses revealed that these variables did not significantly predict category boundary placement for the English contrasts or Spanish /p/–/b/ (English /p/–/b/: $F(3, 15) = 0.199$, $p = 0.896$, $R^2 = 0.038$; English /k/–/g/: $F(3, 15) = 0.569$, $p = 0.644$, $R^2 = 0.102$; Spanish /p/–/b/: $F(3, 15) = 0.193$, $p = 0.900$, $R^2 = 0.037$). As revealed by the $R^2$ values, this set of variables only accounted for 3.8% (in the case of the L1 bilabial contrast), 10.2% (for L1 velars) and 3.7% (for L2 bilabials). In the case of Spanish /k/–/g/, the analysis revealed a significant result ($F(3, 15) = 4.148$, $p = 0.025$, $R^2 = 0.453$). Specifically, in this case, only L2 instruction was found to add statistical significance to the prediction ($p = 0.00$). Note that, in spite of the significant result, the independent variables only accounted for 45.3% of variability.

*3.2. Interim Discussion*

The L1 and L2 perception of bilabial and velar stops by English learners of Spanish was analyzed by means of a forced-choice categorization test for each place of articulation involving stimuli from a VOT continuum. The values of the category boundaries obtained by the English learners of Spanish were considerably late (16.1 for English /p/–/b/; 28.4 for English /k/–/g/; 15.7 for Spanish /p/–/b/ and 26.7 for Spanish /k/–/g/), which suggests that they presented category boundaries for voiced and voiceless velar and bilabial stops that matched the English values. That is, aspiration was needed in order to perceive a voiceless category. In line with these results, Gorba (2020) and Gorba and Cebrian (2021), who also included monolingual speakers in the analysis, reported that the English learners obtained English-like values for both English and Spanish; significant differences were observed between all L2 learner groups and the Spanish monolinguals, but no differences between the L2 learner groups and the English monolinguals were found.

Moreover, in this study, no effect of FTE on the perceived category boundary between voiced and voiceless stops was found. This could be explained by the small range in individual FTE values and the fact that the values were rather low. Recall that FTE is calculated by using length of residence and the proportion of L2 used. Participants had spent between 4 months and 5 years in an L2 setting and the FTE values ranged from 0 to 1.9. According to Flege (2021), who tested the effect of FTE on stop production, 2 years of FTE are necessary to result in notable improvements in accuracy. The present paper seems to be in line with the findings for production and it also suggests that less than 2 years of FTE cannot predict L2 accuracy. In fact, previous studies that found a clear effect of LOR on both the L1 and the L2 focused on participants who had spent a longer period of time in an

L2 setting (e.g., Flege 1987, with a LOR of 12 years). As for language use, the second factor used to calculate FTE, none of the participants used their L2 to a greater extent than their L1. Half of the participants were living in Spain at the time of testing, whereas others were already back in their home country. Despite this difference, the use of the L2 was rather limited across all participants. This is probably related to the fact that the testing for those living in an L2 setting took place in Barcelona, a city with a large international community, which may have favored the use of English, and that participants used mostly English at work. The English-like values of the L1 category boundaries—i.e., they were not drifting towards L2 values—can be explained by the fact that using the L1 even in an L2 context preserves the L1 from L2 influence (e.g., Carlson 2018; Tobin et al. 2017). In fact, L1 use, particularly in a professional context, has been reported to be the strongest predictor of L1 stability (Schmid and Dusseldorp 2010; Schmid and Jarvis 2014). To further explore the role of LOR, L2 use and L2 instruction, the degree to which these factors predicted category boundary was assessed for each contrast and language. A multiple regression analysis revealed that only L2 instruction significantly predicted category boundary placement in the case of /k/–/g/ boundary in Spanish. This result suggests that L2 instruction can improve the perception of L2 stops, at least in the case of velars, but the inconsistency across place of articulation suggests that other variables, which may not have been analyzed in the current paper, may also play a role. As a matter of fact, the multiple regression analyses conducted for the English contrasts and for Spanish /p/–/b/ were non-significant, indicating that the factors included in the analyses did not significantly predict the results. Overall, the variables investigated in the current paper account only for a relatively small percentage of the variance in category boundary placement, which points to the need to investigate additional factors.

Another issue that needs to be addressed is the lack of a clear difference between L1 and L2 category boundaries. As has been mentioned, similar results were obtained for both English and Spanish categories, and these were English-like. This result could be explained, as mentioned above, by a short LOR and rather low use of the L2, but also by potential methodological limitations. The perception tasks that the participants had to complete presented the same stimuli to test both languages—that is, prevoiced, short-lag and aspirated stimuli were used in both tests. Being exposed to aspirated stops, a feature present in English but not in Spanish, it is possible that the English learners activated an English mode in the Spanish perception task (Grosjean 2001). All in all, no clear effect of FTE was observed in the category boundary placement of the /p/–/b/ and /k/–/g/ contrasts in English or Spanish on the part of English learners of Spanish. Moreover, the learners seemed to perceive both languages using English-like values, which suggests an effect of the L1 on L2 perception, but no effect of the L2 on the L1. Next, the results obtained for the Spanish participants will be presented.

## 4. Spanish Learners of English

### 4.1. Results

The same procedure was conducted for the Spanish learners. Table 4 presents the category boundaries and FTE for each Spanish participant. The mean FTE calculated for the Spanish learners of English was 2 (SD: 2.2) and their range was greater than in the case of the English learners of Spanish (0–7.4). However, most participants—i.e., ten—presented low FTE values, from 0 to 0.3, whereas five of them presented values between 0.4 and 2—i.e., medium values, according to Flege (2021)—and three of them presented higher FTE values which were above 2 (see Table 4 and Figures 3 and 4 for individual results). The mean category boundaries were later in English than they were for the Spanish contrasts: English /p/–/b/: 11 (SD: 1.8); English /k/–/g/: 24.6 (SD: 3.3); Spanish /p/–/b/: 9.1 (SD: 3.6); Spanish /k/–/g/: 20.9 (SD: 2.9). None of the tests yielded a significant correlation (see Table 5). Moreover, as the scatterplots in Figures 3 and 4 show, the examination of individual values did not show a clear pattern, as participants with the highest FTE

values did not always present later category boundaries than the mean, whereas that was sometimes the case for participants with low FTE values.

**Table 4.** Percentage of L2 use, years spent in an L2 setting, FTE and the category boundaries for the bilabial and velar contrasts in English and Spanish by the Spanish learners of English.

| Participant | % Weekly L2 Use | Years in L2 Setting | FTE | English /p/–/b/ Boundary | English /k/–/g/ Boundary | Spanish /p/–/b/ Boundary | Spanish /k/–/g/ Boundary |
|---|---|---|---|---|---|---|---|
| Sp01 | 53% | 2.1 | 1.1 | 13.9 | 24.3 | 9.7 | 21.8 |
| Sp02 | 57% | 1.3 | 0.8 | 15.1 | 22.3 | 14.2 | 26.3 |
| Sp03 | 47% | 3.2 | 1.5 | 9.3 | 22.0 | 9.6 | 21.5 |
| Sp04 | 40% | 3.2 | 1.3 | 9.0 | 28.2 | 3.9 | 14.0 |
| Sp05 | 93% | 8.0 | 7.4 | 10.3 | 26.6 | 9.3 | 22.3 |
| Sp06 | 87% | 4.2 | 3.6 | 11.3 | 22.0 | 13.5 | 23.8 |
| Sp07 | 20% | 4.0 | 0.8 | 11.7 | 26.3 | 11.4 | 20.2 |
| Sp08 | 40% | 5.0 | 2.0 | 16.8 | 22.3 | 15.2 | 18.0 |
| Sp09 | 27% | 0.6 | 0.2 | 15.2 | 33.3 | 7.7 | 20.2 |
| Sp10 | 10% | 0.3 | 0.0 | 15.8 | 28.8 | 0.4 | 21.9 |
| Sp11 | 40% | 0.5 | 0.2 | 9.7 | 22.4 | 10.0 | 20.2 |
| Sp12 | 17% | 0.4 | 0.1 | 9.0 | 22.7 | 11.7 | 21.6 |
| Sp13 | 13% | 0.5 | 0.1 | 7.7 | 24.8 | 7.7 | 24.5 |
| Sp14 | 27% | 0.5 | 0.1 | 8.0 | 22.6 | 7.4 | 18.4 |
| Sp15 | 37% | 0.7 | 0.3 | 15.8 | 26.3 | 10.0 | 17.8 |
| Sp16 | 30% | 0.4 | 0.1 | 8.0 | 25.8 | 8.0 | 24.1 |
| Sp17 | 30% | 0.3 | 0.1 | 8.0 | 20.2 | 10.0 | 20.2 |
| Sp18 | 33% | 1.0 | 0.3 | 3.4 | 22.0 | 4.9 | 19.6 |

**Table 5.** Results for the correlation tests between FTE and category boundaries for the Spanish learners of English.

| | | English /p/–/b/ | English /k/–/g/ | Spanish /p/–/b/ | Spanish /k/–/g/ |
|---|---|---|---|---|---|
| FTE | Pearson Correlation | 0.061 | 0.007 | 0.234 | 0.117 |
| | Sig. (1-tailed) | $p = 0.405$ | $p = 0.490$ | $p = 0.165$ | $p = 0.322$ |
| | $N$ | 18 | 18 | 18 | 18 |

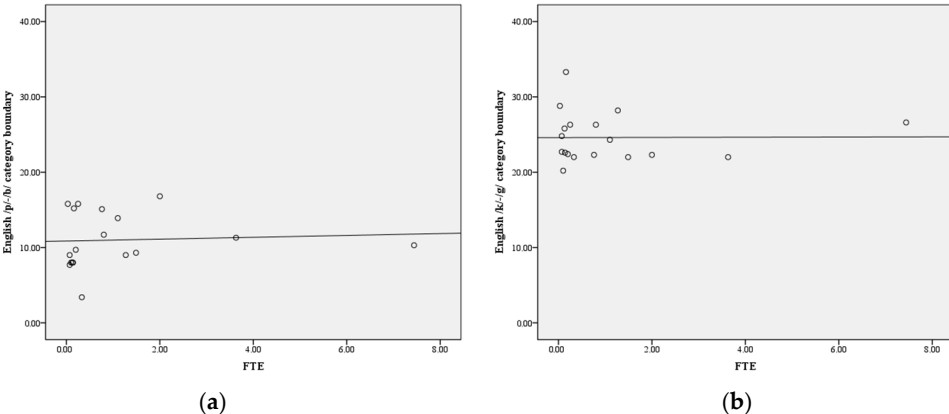

(**a**)        (**b**)

**Figure 3.** Scatterplots with fit lines for FTE (*x* axis) and category boundary (*y* axis) in English by the Spanish learners of English. (**a**) Category boundary for the /p/–/b/ contrast in English. (**b**) Category boundary for the /k/–/g/ contrast in English.

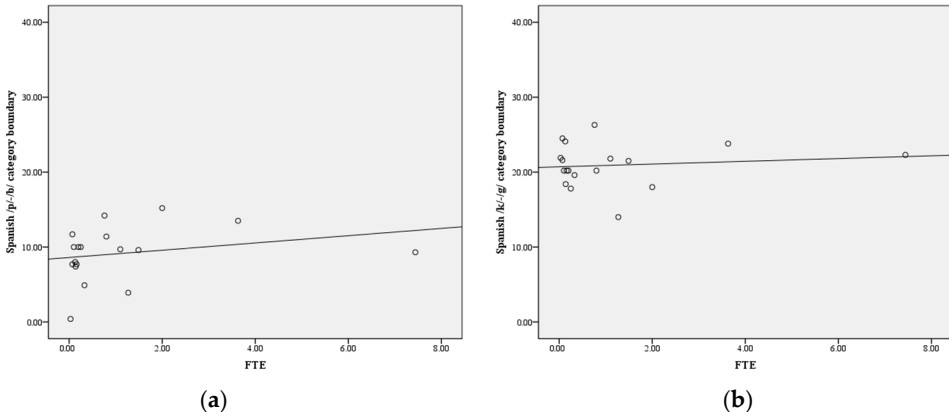

**Figure 4.** Scatterplots with fit lines for FTE (*x* axis) and category boundary (*y* axis) in Spanish by the Spanish learners of English. (**a**) Category boundary for the /p/–/b/ contrast in Spanish. (**b**) Category boundary for the /k/–/g/ contrast in Spanish.

　　　Just as in the case of the English learners of Spanish, a multiple regression analysis was conducted for each language and contrast using standardized Z-scores for each independent variable, with a view to determining to what extent each of the factors investigated predicted category boundary location. None of the tests were found to significantly predict category boundary placement (English /p/–/b/: $F(3, 14) = 0.113$, $p = 0.951$, $R^2 = 0.024$; English /k/–/g/: $F(3, 14) = 1.064$, $p = 0.396$, $R^2 = 0.186$; Spanish /p/–/b/: $F(3, 14) = 1.751$, $p = 0.203$, $R^2 = 0.273$; Spanish /k/–/g/: $F(3, 14) = 0.772$, $p = 0.529$, $R^2 = 0.186$). In fact, as the $R^2$ value indicated, this set of variables accounted only for a small percentage of variability in category boundary placement (2.4%, in the case of the English bilabial contrast; 18.6%, for English velars; 27.3% for Spanish bilabials and 18.6% for Spanish velars).

*4.2. Interim Discussion*

　　　The Spanish learners of English completed the same tasks as the English learners of Spanish, namely a categorization task involving /p/–/b/ and another with /k/–/g/ for each language. The category boundaries obtained by the learners were similar in both languages, though somewhat earlier in Spanish (Spanish /p/–/b/: 9.3; Spanish /k/–/g/: 22.3) than in English (English /p/–/b/: 10.3; English /k/–/g/: 26.6), and earlier than those of the English speakers. Gorba (2019, 2020) also tested Spanish monolingual speakers, who obtained values of 3.4 for /p/–/b/ and 14.8 for /k/–/g/. In other words, the Spanish learners of English presented intermediate values between Spanish and English. Gorba (2020) established that the group with the longest LOR presented the latest—most English-like—values for both languages. In the current paper, the results were reanalyzed to determine if FTE was related to the placement of category boundary in both the L1 and the L2 as well as to assess the degree to which L2 use, LOR and L2 instruction predicted category boundary placement. In order to test this, FTE was correlated for each language, and a series of multiple regressions including LOR, L2 use and L2 instruction were carried out. The results showed that these variables did not significantly account for the variability in category boundary placement.

　　　As for the difference between the L1 and the L2, the results in Gorba (2020) showed that, overall, Spanish learners perceived L1 and L2 stops similarly, although it was observed that there was a numerical difference in the expected direction—i.e., later category boundaries were used in English than in Spanish. It has been argued that categories were merged given their similar values in both languages (Flege 1995). Moreover, the group with the greatest experience had the latest category boundaries for both languages, which suggested that having experience of living in an L2 setting resulted in more English-like category boundaries, not only in the L2 but also in the L1. Thus, some instances of phonetic drift were observed in the Gorba (2020) study. However, the results in this paper do not show a straightforward relationship between FTE and category boundary placement.

Specifically, as mentioned above, no significant correlations were found, and, moreover, individual values showed that participants with a greater FTE did not always obtain category boundaries greater than the mean—i.e., more English-like. Moreover, LOR, L2 use and L2 instruction were not found to significantly predict category boundary placement.

The lack of a clear effect of FTE on the perception of stops may be related to methodological issues, which could involve other factors, such as the quality of the input received, amount of L2 instruction and sample size, as will be explained in the general discussion. It could also be related, as has already been mentioned, to the fact that the LOR of the participants was relatively short (*M* = 2 years). In sum, it appears that FTE did not predict the location of the /p/–/b/ and /k/–/g/ category boundaries in the L1 and L2 of Spanish learners of English, as no significant results were observed. In short, the four variables tested (FTE, LOR, L2 use and L2 instruction) did not seem to clearly predict or correlate with the perception results, suggesting that more variables should be considered to assess L2 stop category-boundary placement and the effect of L2 acquisition on L1 category boundary placement.

## 5. General Discussion

In the present paper, a reanalysis of the perceptual data in Gorba (2020) and Gorba and Cebrian (2021) was conducted by calculating the FTE of those participants who had lived in an L2 setting and correlating it with the category boundaries they obtained in each language. The correlations were very weak for both populations, and none of them were significant. Moreover, an individual examination of the data did not reveal any consistent pattern across participants—that is, participants with greater FTE values did not always obtain more L2-like category boundaries. It should be noted that the dataset in the current paper was small, as only the category boundaries of 19 or 18 participants were used in each analysis, and that the insufficient range in individual FTE values, particularly in the case of the English population, may have prevented the establishment of true relationships between FTE and category boundary. Furthermore, a series of multiple regression analyses involving LOR, L2 use and years of L2 instruction were also conducted. The overall results showed that these variables did not significantly predict category boundary placement. However, amount of L2 instruction seemed to account for category boundary placement in the L2 to a greater extent in the case of the English learners of Spanish.

The results in Gorba (2020) and Gorba and Cebrian (2021), which used a different type of analysis, suggested a more straightforward relationship between /p/–/b/ and /k/–/g/ category boundaries and group, which was assigned according to LOR, than FTE did in the current paper. This result may seem surprising since FTE also involves LOR and, moreover, incorporates an extra factor, L2 use. One possible explanation for this outcome is related to the nature of the variables used in each analysis. In the case of Gorba (2020), LOR was treated as a categorical variable, and L2 learners were classified in three groups accordingly. Specifically, there was a group of L2 speakers who had been living in the L2 setting for four years on average, a group of L2 speakers who had lived in an L2 setting for 7 months in the case of the Spanish speakers and 9 months in the case of the English speakers and were back in their home country at the time of testing, and a group that had only received instructional learning, which is not included in the current study. Here, the main factor, FTE, was instead used as a continuous variable. Moreover, the FTE values obtained were generally low, and, thus, small differences were observed between participants, particularly in the case of the L1-English speakers (Spanish participants: Mean: 2, SD: 2.2; English participants: Mean: 0.4, SD: 0.6). In fact, Flege (2021) found that at least 2 years of FTE were necessary for Japanese learners of English to perceive the differences between L1 and L2 liquids. Moreover, the SLM-r deemed the amount of FTE received by the experienced L1-Mandarin and L1-Spanish learners of English in Flege et al. (1992), who obtained an FTE value of 4.2, as inadequate in order to produce /t/ and /d/ with a native-like degree of intelligibility. In the current study, most participants presented lower values (Spanish speakers' mean: 2.2; English speakers' mean: 0.2). Nonetheless, more straightforward

results were expected in the case of the Spanish speakers, who presented a greater FTE range (0–7.4) than the English participants (0–1.9). In the case of the English learners of Spanish, the null result could actually be explained by the very short FTE range across participants. In this regard, the SLM-r claims that category formation is a slow and gradual process. Thus, it is possible that the English learners of Spanish had not received sufficient input and, as a result, did not present high enough FTE values. However, previous studies found evidence of great improvement in relatively inexperienced L2 learners with just short periods of intensive L2 use (Casillas 2019; Casillas and Simonet 2018; Kartushina and Martin 2019), even in an instructional setting. This suggests that rapid changes can occur with a short period of exposure to the L2 and, thus, low FTE values. Moreover, Casillas and Simonet's work points to the importance of the input received in instructional settings, not only in immersion settings.

It should also be acknowledged that participants differed in other factors, particularly regarding L2 instruction. More specifically, the Spanish learners of English had learned the L2 for a greater period of time than the English speakers (14 vs. 5 years, respectively), as English is mandatory in the curricula of Spanish primary and secondary schools. Moreover, the L1-English participants who had lived in an L2 setting and were back in their home country were majoring in Spanish, the L2, whereas that was not the case for the English teachers living in the L2 setting, who had received less formal instruction (7.5 years vs. 2.4 years, respectively). It is possible that, although the subset of participants living in an L2 setting presented longer LORs and, as a result, greater FTE values (0.12 vs. 2.04, respectively) than the subset living in the UK, the fact that they had received less L2 instruction levelled them up with those with a shorter—and past—stay but with a greater amount of L2 instruction. In fact, a greater amount of L2 instruction was found to be positively correlated with target-like /k/–/g/ category boundaries.

Furthermore, in this study, all participants with experience of living in an L2 setting were included in the same analysis. Thus, no difference was made regarding the place of testing. It is possible that, given that LOR was not particularly long for any of the participants, the greatest differences in performance involved those participants who had had the experience of living in an L2 setting as opposed to those who had not. Therefore, it is plausible that the relatively small differences between participants and the small dataset—18 or 19 participants in each analysis—resulted in weak results. Another possible explanation could be related to the recency of the stay, since previous studies show differences in this regard (e.g., Sancier and Fowler 1997). In particular, a more recent stay in an L2 setting has been found to result in a more L2-like performance in both languages. Moreover, according to Grosjean (2001), setting facilitates the activation of the language spoken there. Hence, living in an L2 setting would be expected to result in more L2-like category boundaries in both languages. Nevertheless, that was not always the case in the present study. In fact, it was the participants with the greatest LOR who were living in an L2 setting; thus, setting and recency would be expected to only emphasize the effect that was expected for FTE—i.e., result in a more L2-like performance. It should also be noted that, as pointed out by previous research (Chang 2012, 2013), a longer length of residence may not necessarily result in more L2-like productions in the L1, as L2 learners with no previous contact with the L2 may present greater L1 drift towards L2 values due to a novelty effect that may fade as they become more experienced in the target language.

Other factors that were not analyzed in this study could have affected the L2 and L1 of the L2 learners, such as inhibitory skill (e.g., Darcy et al. 2014; Lev-Ari and Peperkamp 2013). Specifically, lower inhibitory skill may result in greater influence, from both the L1 on the L2 and the L2 on the L1. It should also be noted that cognitive control may also depend on the context. Beatty-Martínez et al. (2020) found that Spanish–English bilinguals with lower proactive control were better able to retrieve the L2 in an L2 setting, but at the expense of the L1, whereas those with higher proactive control maintained access to the L1. Thus, it is possible that there were differences, in terms of inhibiting the language, between participants in an L1 setting as opposed to those in an L2 setting that were not being tested.

Cognitive variables, such as phonological working memory (Perrachione et al. 2017) and short-term memory, which were not tested in the current study, also seem to play a role on L2 category formation formation (Aliaga-Garcia et al. 2011) and cue weighting (Cerviño-Povedano and Mora 2015). Attitudinal variables may also influence the L2 and the L1, although less clear results have been found in the literature (e.g., Schmid and Dusseldorp 2010). In fact, conflicting results have been reported when it comes to motivation (e.g., Moyer 1999; c.f., Thompson 1991). Some studies determined that, even though it had some effect on accentedness, its effect was comparatively smaller than in the case of other individual factors (e.g., Flege et al. 1999). In short, attitudinal variables, which were not assessed in the current paper, could have influenced the participants' performance, although, according to previous studies, their effect might have been relatively small. Another factor that could have influenced the results is phonological awareness which, in this case, would involve the specific implementation of VOT in the languages under study. Metalinguistic awareness, which refers to explicit knowledge about language and which varies across learners (Kivistö-de Souza 2015), could have had an effect on the performance of L2 learners, as reported by previous studies (De Leeuw et al. 2019; Kivistö-de Souza 2015). No tests assessing phonological awareness were conducted in this study. Awareness may have also risen through instruction. Although it is possible that pronunciation was addressed in class, no information regarding explicit teaching or feedback on the part of the teachers was gathered. Another factor that needs to be considered is the type of communicative setting, as L2 speakers in contact with their L1 through settings in which code-mixing is inhibited have been found to present greater L1 stability (De Leeuw et al. 2010). Considering all the possible individual factors discussed above, further research should involve more variables or compound variables that take into account as many factors as possible in order to be able to assess and determine which factors contribute—and to what extent—to L2 speech acquisition and crosslinguistic influence.

Finally, the current paper only considered VOT, the main cue for voicing in stops in English and in Spanish (e.g., Lisker and Abramson 1964). In fact, as explained in the methods, secondary cues were controlled by using ambiguous values—i.e., between voiced and voiceless values—so as to ensure reliance on VOT. It would be interesting to see whether similar results were obtained using secondary cues, such as F0.

Overall, the outcome of the current paper does not show that FTE is an appropriate factor to assess the data presented in this study. In other words, FTE did not predict CLI in stop perception assessed via category boundary measurements when it comes to L2 speakers with low and mid L2 input. In fact, no significant correlations were observed between FTE and L2 and L1 performance, and no clear patterns were revealed by individual results. Although some methodological limitations could explain this outcome, the results of the paper suggest that FTE seems more suitable for investigating highly experienced participants or for use with datasets in which a wider range of FTE values are represented. This paper also emphasizes the need to not only quantify the amount of L2 use but also determine the quality of the input, and it reminds us that other factors, including L2 instruction, also play an important role in L2 speech. In fact, the four variables that were tested did not successfully predict the results. Thus, a more comprehensive variable or set of variables needs to be taken into consideration, as L2 instruction, metalinguistic awareness, context, and cognitive and attitudinal factors can also influence L2 acquisition as well as L1 drift and attrition.

**Funding:** This research was funded by Research Grants No. FFI2017-88016-P and PID2021-122396NB-I00 from the Spanish Ministries of Economy and Competitiveness and Science and Innovation. The APC was funded by Universitat Autònoma de Barcelona.

**Institutional Review Board Statement:** Ethical review and approval were not required for this study in accordance with the local legislation and institutional requirements.

**Informed Consent Statement:** Informed consent was obtained from all subjects involved in the study.

**Data Availability Statement:** Not applicable.

**Conflicts of Interest:** The authors declare no conflict of interest. The funders had no role in the design of the study; in the collection, analyses, or interpretation of data; in the writing of the manuscript, or in the decision to publish the results.

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
