# Peer review of "Is Full-Time Equivalent an Appropriate Measure to Assess L1 and L2 Perception of L2 Speakers with Limited L2 Experience?"

_languages, doi:10.3390/languages8010056_

Round 1

Reviewer 1 Report

Please see attached

Author Response

Dear Reviewer,

Thank you very much for having taken time to review my paper and for your very insightful comments. I reply to each of them below:

 Review of Is Full Time Equivalent an appropriate measure to assess L1 and L2 perception?

This is a nice use of an existing data set to examine the hypothesis that FTE correlates with L2 category development and, with some expansion of the framing and discussion can make a solid contribution to the empirical examination of the SLM-r. Most importantly, as detailed below, given that FTE was not shown to correlate with category boundaries, more robust discussion of what other individual differences should be pursued as explanatory (based on existing research in L2 perception and L1 attrition) and the role of VOT as the test case, will strengthen the paper.

Introduction/background

  1. 2 When discussing the previous literature on L2 stops, although most readers will be familiar with VOT and category boundaries and there is an overview in Section 1.1, it would still be helpful to note that Spanish is a voicing language and English an aspirating language and how their category boundary patterns are reported to differ.

Thanks for pointing this out. A reference to VOT use in the languages under study has been made where suggested.

  1. 3 “Flege et al. (2021) determined that at least 2 years of FTE were needed”. FTE at what value? That is, what percentage of L2 usage would be necessary?

FTE is calculated by combining length or residence and L2 use. Thus, years living in an L2 setting does not equal years of FTE. The authors do not provide information about the minimum L2 use needed but talk in terms of FTE. The formula to calculate this is specified in the text and more details about the participants in the study referred to in Flege et al. (2021), including L2 use, have been provided.

  1. 2 Cabrelli et al. (2019) looked at perception and I’m not sure that changes in illusory vowel perception are best categorized as ‘less accurate’.

This has been rephrased.

Methods and results

In the description of the participants, please include the standard deviations for the mean LoR, percentage of L2 usage, and number of years, so as to inform the degree of variability in the sample.

Done.

For learners living in Barcelona, how was Catalan exposure evaluated? Were any of the L2ers proficient in Catalan? What was their comprehension proficiency? This could be a factor since Catalan VOT patterns with that of Spanish and would therefore essentially increase their FTE.

They did not know or use Catalan. This has been specified in the manuscript.

What about recency? This is a confounding variable since the shorter-term residents also happen to be the ones that are not currently in an L2 environment. How would one know if any observed effects are due to FTE or to recency?

Thanks for bringing this up. This is definitively a variable that needs to be taken into consideration. This was already included in the discussion but has been made clearer in the revised version of the manuscript.

I would like to see separate correlations that isolate LoR and L2 usage to see if it provides any additional insight.

The correlations have been conducted and are now reported in the manuscript. Its results are also addressed in the discussion.

Please include more information about the bilingual speaker that recorded the stimuli. What dialect of each language did they speak? Were they English-dominant or Spanish-dominant? Early or late learner?

The speaker was a bilingual native speaker of Spanish and Catalan (specifically, standard European Spanish and Central Catalan, two languages that do not differ in either VOT production of initial stops), who was a trained phonetician, had started learning EFL in his childhood and had spent about seven years in North America/English-speaking Canada as an adult. Moreover, recall that the stimuli were manually manipulated to obtain the desired VOT values. That is, the recordings were not used as they were obtained, but were edited for the creation of the continua.

What software was used for stimuli presentation and data recording?

Praat was used for both. This has been specified in the manuscript.

It would be helpful to put the category boundary data in a table.

I am not sure I understand this suggestion. The category boundaries for each place of articulation and language are provided in Tables 2 and 4.

Findings and discussion

Even considering the size of the data set and the shorter LoRs in the sample, I don’t know that I find it surprising that FTE does not correlate with the category boundaries. There is no doubt that FTE is relevant, but there are a number of other variables at play in sources of individual variation in perception (working memory, retrieval-induced inhibitory control, phonological short-term memory), not to mention social-affective variables such as motivation in ultimate attainment. Another thing to consider is the difference between bilinguals in an L1 context versus an L2 context. For example, recent research from Beatty-Martínez et al. (2020) has shown that cognitive control varies as a function of whether a bilingual is in a context in which the language that will be used (L1 or L2) is more predictable or more varied. I think that making these other variables a focus of discussion in terms of future steps (with perhaps the suggestion for a study that examines a number of these variables and uses principle components analysis would enhance the discussion and the overall impact of the paper (see e.g., Schmid & Dusseldorp, 2010, for an illustration of how this exploratory analysis has been used in L1 attrition research).

Thank you very much for suggesting these interesting references. The discussion has been developed following the reviewer’s suggestion.

  1. 13 I would note here that the finding is in line with attrition research that has shown that L1 usage in a professional context has been found to be the strongest predictor of stable L1 representations (albeit for lexical access, see Schmid & Jarvis, 2014).

Thank you for pointing this out. This could explain the stability of the L1 of the English learners as well as the drift towards the L2 as a function of L2 use that the new correlations show in the case of the Spanish learners of English.

  1. 14 Regarding the potential effect of metalinguistic awareness, I would include any existing research about MA and VOT specifically. This is something that is not instructed in an L2 Spanish classroom unless the learners take a phonetics/phonology class.

It is true that L2 instruction does not necessarily involve pronunciation teaching and that that is actually not often the case. However, some teachers do teach pronunciation, either through explicit instruction or by giving students feedback. Previous studies have pointed out that instruction may improve awareness, regardless of the type (i.e., phonetic instruction vs. implicit teaching, Kissling 2013). This has been acknoeldged in the general discussion.

It might be nice to include a brief discussion of the use of VOT as a test case. As we know, Flege et al would consider these to be ‘similar’ categories and L2ers might not be attending to VOT as a cue in the input. It would be interesting to replicate this with phenomena of varying salience.

I understand your point. However, as pointed out in the methodology, the stimuli controlled for secondary voicing cues, including burst intensity and duration, F0 and F1, by making them ambiguous (i.e., with intermediate values between voiced and voiceless). The intention was to make sure that participants were relying solely on VOT. Previous studies have in fact investigated the effect of other cues, such as F0, and determined that they were only decisive when the listener was represented with stimuli with ambiguous VOT values and have been associated with a bias towards voiceless stops (Idemaru and Holt 2011; Whalen, Abramson, Lisker and Mody 1993). I believe that this possibility has been left out by neutralizing secondary cues. Still, you point has been acknowledged in the manuscript.

  1. 14 “This paper also emphasizes the need to not only quantify the amount of L2 use but also to determine the quality of the input”. I think it emphasizes the need to look at other variables, period, as mentioned above. The discussion of quality of input in terms of percentage of accented input is mentioned in the beginning but is not addressed again until the conclusion so it comes across as a bit of an afterthought. Based on its mention in the first half of the paper, I was surprised that input quality was not incorporated in the FTE calculation, so I would suggest maybe not including it in the first part and instead bringing it up along with the other variables in the discussion.

Unfortunately, I did not have data on the quality of the input. Moreover, Flege, Aoyama and Bohn (2021) calculate FTE in the same way as the current paper does for the same reason, although the authors emphasize its importance in the discussion. Following your suggestion, references to quality before the discussion have been eliminated.

Copyedits:

  1. 1 Please spell out acronyms (e.g., SLM and PAM) at first mention.
  2. 2 “on the perception L1 stops” --> on the perception of L1 stops
  3. 3 perceviced --> perceived
  4. 4 Weenik --> Weenink
  5. 5 beeltle --> beetle
  6. 8 “In line with this results” → these
  7. 8 “English learner of Spanish” --> learners
  8. 8 “none of the test” → tests
  9. 8 “by deflecting from one another” --> via dissimilation
  10. 13 “for neither population” --> either

The typos above have been corrected

Reviewer 2 Report

The submission presents a re-analysis of native and nonnative perception of stimuli from a VOT continuum, using as independent variable the Full Time Equivalent (FTE), which Flege & Bohn (2021) suggest should replace Length of residence (LOR) as a measure of L2 experience. The main result of the study is that FTE is not significantly related to the VOT boundaries (in two places of articulation) of native Spanish and native English listeners when they perceive stimuli from this continuum in an assumed ”Spanish” or “English” mode.

The main problem with the submission is not that it presents a null finding – I do not believe that only positive findings should be reported. The problem is that the data used by the author cannot address the question of whether FTE is a valid measure of L2 experience because, as the author acknowledges, this range is very narrow. Only two of the 19 L1 English listeners have an FTE of > 1.0, and only six of the 18 L1 Spanish participants have an FTE of > 1.0. The overall conclusion (p. 14, 539) that “the outcome of the present paper does not support the idea that FTE is a better factor to assess CLI in stop perception than LOR” must be modified to include the important reservation “for very low FTEs”. I suggest, therefore to frame the research question addressed on the submission more precisely: “Is Full Time Equivalent an appropriate measure to assess L1 and L2 perception for learners with limited L2 experience?“  - In general, I suggest that the author refrains from mentioning (in quite some detail) effects of FTE that or nonsignificant, such as on p. 10, 379, p. 12 427.

Some passages may need correction:

p. 3, 73, I do not understand  “there was an influence of the L2 on the L1 and no effect of the l2 was observed on the L1”. Isn’t that contradictory?

p. 3, 128: In what sense does Spanish present a shorter VOT range? The difference between phonologically voiced and voiceless stop is clearly larger (in ms) in Spanish than in English.

p. 5, 215: “percentage” should be “proportion”

p.7, 248: Is this a typo? Shouldn’t it be the case that high FTE values move the VOT boundaries towards Spanish values?

Author Response

Dear reviewer,

Thank you very much for you feedback and for having taken the time to review my manuscript. I have answered your comments below:

The main problem with the submission is not that it presents a null finding – I do not believe that only positive findings should be reported. The problem is that the data used by the author cannot address the question of whether FTE is a valid measure of L2 experience because, as the author acknowledges, this range is very narrow. Only two of the 19 L1 English listeners have an FTE of > 1.0, and only six of the 18 L1 Spanish participants have an FTE of > 1.0. The overall conclusion (p. 14, 539) that “the outcome of the present paper does not support the idea that FTE is a better factor to assess CLI in stop perception than LOR” must be modified to include the important reservation “for very low FTEs”. I suggest, therefore to frame the research question addressed on the submission more precisely: “Is Full Time Equivalent an appropriate measure to assess L1 and L2 perception for learners with limited L2 experience?“  

Thank you very much for your comment. The title, research questions and discussion have been changed accordingly. In fact, more information about FTE and the participants with which Flege and colleagues tested the factor has been provided. The FTE values of the participants in the current study would not be considered to have received low to mid input, as they fall between the values for the groups labelled in this manner in Flege and Bohn (2021). Thus, the goal has been rephrased as ‘to determine whether low to mid FTE values can account for differences in the perception of L2 and L1 stops on two parallel populations’.

- In general, I suggest that the author refrains from mentioning (in quite some detail) effects of FTE that or nonsignificant, such as on p. 10, 379, p. 12 427.

The report of the statistical analysis has been summarized as suggested.

Some passages may need correction:

  1. 3, 73, I do not understand  “there was an influence of the L2 on the L1 and no effect of the l2 was observed on the L1”. Isn’t that contradictory?

This was a typo. It has been corrected.

  1. 3, 128: In what sense does Spanish present a shorter VOT range? The difference between phonologically voiced and voiceless stop is clearly larger (in ms) in Spanish than in English.

English also uses voice-lead in certain contexts and, specifically in initial position, voice-lead VOT may also be used in free variation. That is, even though prevoicing is not used phonologically it does occur in English. Thus, VOT in English ranges from voice-lead to long-lag VOT, whereas, in Spanish, it ranges from voice-lead to short-lag.

  1. 5, 215: “percentage” should be “proportion”

Fixed.

p.7, 248: Is this a typo? Shouldn’t it be the case that high FTE values move the VOT boundaries towards Spanish values?

The reviewer’s assumption points out is what we would expect if FTE resulted in phonetic drift of the L1 towards the L1. Another possibility could go in the direction of dissimilation. Still, as pointed out in the manuscript, in the specific statement this comment is referring to, the predictions concerning the individual analysis were made based on the results of the correlations. That is, it was expected that participants with higher FTE presented later category boundaries in English and earlier in Spanish because the direction of the correlations suggested that – negative vs. positive correlations, respectively. This has been clarified in the manuscript.

Round 2

Reviewer 2 Report

The revised version is clearly improved, but I still have one major concern and one minor comment. The manuscript could be accepted for publication once this major concern is satisfactorily addressed.

The author(s) modified the title, RQs, and discussion to address my criticism of the original submission, which concerned the claim that the data presented could address the question of whether FTE is an appropriate measure of L2 speech learning (for any FTE). The revision is more accurate in specifying the claim with regard to "L2 speakers with low to mid L2 input". I still think that this claim not supported by the data, and I suggest that the author(s) replace their formulation with the wording i suggested, namely, "limited L2 experience" instead of "low to mid L2 input" throughout the manuscript. 

The graphs, especially for the L1 English participants, clearly show that the very few participants which could be considered as having "mid L2 input" are outliers and as such can't be used to derive conclusions about the relation of FTE to the dependent variables. What all graphs clearly show is that FTE is unrelated to the dependent variables for participants with limited ("low") L2 experience. This result is supported by the data, whereas any claim about L2 speakers with "mid" L2 input is not. (I am aware of the fact that Flege et al. used the "mid" label for learners with an FTE range starting at 0.4, but a) that is very generous, b) the graphs in the submission show that the the L2 English learners in the study under consideration cluster with other learners with limited ("low" L2 experience.)

As a minor and easily implemented change, I suggest either condensing or, preferably, deleting the paragraph near the end of the General Discussion (starting with "Other factors ... ", line 644). The contents is largely speculative, contains too many "could haves", and doesn't add to the study because the author(s) can't and shouldn't address all the potential factors listed in this paragraph.

Typos: I caught just two:

Line 404: Should be "correlated with the category boundaries"

Line 641: ad -> and

Last paragraph

ad

catehory

Author Response

Dear reviewer,

Thank you very much for your comments. I understand your point and the amount of L2 experience received by the participants in this study is now consistently referred to as limited. Thank you very much for pointing out the typos, which have also been corrected.

As for your second comment, the paragraph referring to other factors was included in the previous revised version of the manuscript with an aim to address a concern raised by one of the reviewers as well as the editors and it seems to have fulfilled their request.